# Effect of Pressure Conditions in Uterine Decellularization Using Hydrostatic Pressure on Structural Protein Preservation

**DOI:** 10.3390/bioengineering10070814

**Published:** 2023-07-07

**Authors:** Dongzhe Wang, Narintadeach Charoensombut, Kinyoshi Kawabata, Tsuyoshi Kimura, Akio Kishida, Takashi Ushida, Katsuko S. Furukawa

**Affiliations:** 1Department of Mechanical Engineering, School of Engineering, The University of Tokyo, 7-3-1, Hongo, Bunkyo-ku, Tokyo 113-8656, Japan; 2Department of Bioengineering, School of Engineering, The University of Tokyo, 7-3-1, Hongo, Bunkyo-ku, Tokyo 113-8656, Japan; ch.narintadeach@gmail.com; 3Department of Material-Based Medical Engineering, Institute of Biomaterials and Bioengineering, Tokyo Medical and Dental University, 2-3-10, Kanda-Surugadai, Chiyoda-ku, Tokyo 101-0062, Japankishida.mbme@tmd.ac.jp (A.K.)

**Keywords:** uterine regeneration, tissue engineering, decellularized scaffold, hydrostatic pressure

## Abstract

Uterine regeneration using decellularization scaffolds provides a novel treatment for uterine factor infertility. Decellularized scaffolds require maximal removal of cellular components and minimal damage to the extracellular matrix (ECM). Among many decellularization methods, the hydrostatic pressure (HP) method stands out due to its low cytotoxicity and superior ECM preservation compared to the traditional detergent methods. Conventionally, 980 MPa was utilized in HP decellularization, including the first successful implementation of uterine decellularization previously reported by our team. However, structural protein denaturation caused by exceeding pressure led to a limited regeneration outcome in our previous research. This factor urged the study on the effects of pressure conditions in HP methods on decellularized scaffolds. The authors, therefore, fabricated a decellularized uterine scaffold at varying pressure conditions and evaluated the scaffold qualities from the perspective of cell removal and ECM preservation. The results show that by using lower decellularization pressure conditions of 250 MPa, uterine tissue can be decellularized with more preserved structural protein and mechanical properties, which is considered to be promising for decellularized uterine scaffold fabrication applications.

## 1. Introduction

It is reported that 15% of reproductive-aged couples suffer from infertility worldwide [1]. Uterine factor infertility, which results from congenital or acquired uterine malfunction, is estimated to account for 3–5% of the general population [2]. Despite this, the current treatment is limited to adoption and surrogacy, leading to potential ethical issues. Cases of whole uterus transplantation were also reported [3,4], but are challenged by the organ source shortage and immune rejection reactions.

Uterine regeneration using decellularized scaffolds provides a promising solution to this issue. Decellularized scaffolds are extracellular matrix (ECM) extracted from native tissue or organ by erasing resident cells. They are suitable as tissue grafts due to superior biocompatibility and low possibility of provoking immune rejection reactions. To our knowledge, Santoso et al. were the first to report the fabrication of decellularized uterine scaffold (DUS) and achieved successful pregnancy in using a DUS implanted uterus in a rodent [5]. Miyazaki et al. reported a successful whole uterus decellularization via aortic perfusion with detergents and accomplished a partial reconstruction of the rat uterus [6]. Starting from rat models, the trials of uterine decellularization covered species including rabbit [7,8], pig [9], ovine [10], and most recently human [11]. These reports verified the possibility of functional uterine regeneration via the implantation of DUS.

The balance of cell residue elimination and ECM preservation is vital in decellularized scaffold fabrication. The complete removal of resident cell components is necessary to avoid severe immune rejection reactions. ECM provides not only the attachment site and structural support but also molecular and mechanical cues for cells, which maintain cellular activities and organ function [12]. A fibrous networks of collagen and elastin in ECM provides the dominant response to tissue mechanical forces [13]. Owing to the cell–ECM interaction, the consistency of the composition, microstructure, and mechanical properties between decellularized scaffolds and native tissue can facilitate cell migration, differentiation and mediation of tissue repair after implantation [14,15]. Thus, the damage to ECM during the decellularization process is expected to be reduced.

The majority of trials to fabricate DUS focused on decellularization using detergents, including sodium dodecyl sulfate (SDS) and Triton-X [6,7,8,9,10,16,17,18] or a combination of detergents with physical methods such as freeze–thaw [19]. Our team previously compared the first successful implementation of uterine decellularization using hydrostatic pressure (HP) and detergents, highlighting a better ECM preservation in HP-DUS [5]. 

Since Fujisato et al. implemented the first trial on porcine heart valves using a cold isostatic pressing in 2005 [20], scaffolds fabricated via HP have presented comparatively intact extracellular matrix structure and similar mechanical properties to native tissue compared to those using detergent decellularization [5,21], demonstrating excellent ECM preservation ability of HP decellularization. The lack of involvement of cytotoxic chemicals and sterilization accompanied by decellularization also highlight the biocompatibility of HP decellularized scaffolds.

The previously established uterine HP decellularization research method includes two successive steps: (1) HP treatment at a decellularization pressure of 980 MPa (10,000 atm) to disrupt resident cells; and (2) a follow-up wash with DNase I (Deoxyribonuclease I) solution lasting 7 days to remove cell debris. However, this conventional choice of decellularization pressure (980 MPa) directly threatens the ECM preservation quality due to the partial denaturation of the structural protein [22,23]. It could be inferred that decreasing the pressure condition from the conventional value may alleviate protein denaturation during the decellularization procedure.

While lowering the pressure condition can improve ECM preservation, lower pressures are challenged by consequent reductions in HP decellularization efficiency. Our previous report pointed out that the induction of cell death crucially enables DNase I’s access to nucleus DNA [24]. Recent reports have also revealed that moderate HP (150–250 MPa) can significantly decrease the single cell survival rate in cell suspension models [25,26]. This report examined the cell-killing effect using single cells cultured under moderate HP, which only focused on cell viability under moderate HP. The aforementioned reports leave the following aspects unanswered: (1) whether lower HP than conventional 980 MPa can remove DNA from tissue; (2) whether ECM’s composition and structure can be preserved under moderate HP. This means that the effect of decellularization at a lower HP remains unknown. 

In this paper, the authors therefore conducted HP decellularization on rat uterine tissue with a lower decellularization pressure (250 MPa and 500 MPa) than the conventional pressure of 980 MPa and analyzed the decellularized scaffold quality from the perspective of cell removal, ECM structural protein preservation, and mechanical properties. 

## 2. Materials and Methods

### 2.1. Uterine Tissue Extraction

The Animal Experimentation Committee, Faculty of Medicine, University of Tokyo, censored and approved animal experiment plans and protocols under the project “Rat uterine tissue engineering (ID number is P12-113)”, on 16 December 2011. All animals involved in this study were housed and taken care of following the University of Tokyo Animal Care Facility guidelines. The preparation of uterine tissue was aseptically performed with 9-week-old female Sprague Dawley rats (CLEA Japan, Inc., Tokyo, Japan). The rats were sacrificed via anesthesia overdose using an isoflurane inhalation solution (Pfizer, New York, NY, USA). Uterine horns were separated from the ovaries and vagina after death confirmation. Connective and adipose tissue was trimmed off. A rinse with phosphate-buffered saline (PBS) was conducted to remove blood. Extracted uterine horns were then cut into 1.5 cm long pieces and kept on ice until further use.

### 2.2. Decellularization

The decellularization protocol was adopted from our previous report with modifications [5]. Uterine tissue pieces were sealed in polyethylene bags filled with PBS. HP decellularization was performed using a cold isostatic pressurization machine (Dr. Chef; Kobelco, Kobe, Japan). The samples were divided into three groups and separately pressurized to 250/500/980 MPa for 10 min at 30 °C. Notably, the pressuring and de-pressuring speeds were consistent in the three groups, set as 65.3 MPa/min. After high hydrostatic pressurization, all samples were transferred into a container filled with washing buffer containing 0.9% NaCl (Wako, Tokyo, Japan), 0.05 M magnesium chloride hexahydrate (Wako, Tokyo, Japan), 0.2 mg/mL DNase I (Roche, Basel, Switzerland) and 1% penicillin and streptomycin (Gibco, New York, NY, USA). The samples immersed in the washing buffer were gently shaken at a frequency of 1 Hz and kept at 4 °C in a dark environment for 7 days. The washing buffer was changed daily in a clean environment to prevent contamination. After the 7-day wash, the samples were gently rinsed with PBS. These samples are referred to as DUS samples in the following sections. The entire experimental scheme is shown in Figure 1.

### 2.3. Protein Extraction and Biochemical Analysis of DUS

The DUS samples were freeze-dried for a minimum of 12 h using a vacuum freeze dryer (FDU1200 Eyela, Tokyo, Japan). The weights of freeze-dried samples were measured before the samples were digested in a lysate buffer containing 446 mg/mL of papain (Sigma, St. Louis, MO, USA), 5 mM cysteine-HCl, and 5 mM EDTA-2 Na, at 60 °C for a minimum of 15 h. Papain-digested samples were further refined using a homogenizer and an ultrasonic cell disruptor, and then stored at −80 °C until further use.

DNA contents were quantified via a commercial DNA assay kit (Quant-iT PicoGreen dsDNA assay kit; Invitrogen, Waltham, MA, USA), following the provided protocol. Briefly, the samples and lambda DNA standard were first diluted in Tris-EDTA buffer (Invitrogen, USA). An equal amount of diluted PicoGreen dye was mixed thoroughly with the samples and standards. After 5 min of reaction, the samples and standards were loaded onto a 96-well dish and measured using a spectrophotometer (PerkinElmer, Waltham, MA, USA) at an absorbance wavelength of 522 nm.

The collagen contents were quantified using the hydroxyproline assay, following a standard protocol. Briefly, the samples and hydroxyproline standard were first hydrolyzed using 4M NaOH at 120 °C for one hour, followed by neutralization using 1.4 N citric acid and reaction with Chloramine-T. After 20 min of reaction, the sample solution was mixed with perchloric acid diluted in aldehyde solution and incubated at 70 °C for 60 min. After incubation, the product liquid was transferred onto a 96-well dish. The hydroxyproline contents in each sample were then measured using a spectrophotometer (PerkinElmer, USA) at an absorbance wavelength of 450 nm.

A commercial elastin assay kit (Fastin Assay; Biocolor, Carrickfergus, Northern Ireland, UK) was performed to quantify elastin amounts, following the provided protocol. Briefly, samples and alpha elastin standards were extracted using 0.25M oxalic acid at 100 °C until the weight of unsolvable precipitate stabilized to a constant value. Alpha elastin dissolved in the extracted liquid was precipitated using the provided elastin precipitating reagent. The precipitated elastin contents were pelleted using a centrifuge at 10,000× *g* and dissolved again using the provided dye reagent after discarding the supernatant. Thorough mixing and a 10 min reaction allowed the elastin contents to bind with the dye. Unbound dye was discarded completely after elastin–dye compounds were pelleted via centrifugation at 10,000× *g*. Dried elastin–dye compounds were further dissolved using the dye dissociation reagent provided. The dissolved dye, representing the amount of alpha elastin, was measured using a spectrophotometer (PerkinElmer, USA) at an absorbance wavelength of 513 nm.

### 2.4. Histological Analysis

DUS samples were briefly rinsed with PBS and fixed in the 10% neutral buffered formalin solution overnight. The samples were then dehydrated with a 30% sucrose solution for a minimum of 2 h and embedded in an OCT compound (Tissue-Tek: 4583, Sakura Finetek USA, Inc., Torrance, CA, USA). The sample blocks were frozen in liquid nitrogen and sliced into 5 μm sections using a cryostat (Leica CM1850, Watzlar, Germany).

For histology, these specimens were stained with DAPI (Dojindo, Kumamoto, Japan) to visualize the DNA residue. Briefly, the samples were immersed in running tap water to remove the embedding medium. After 3 min of cell membrane penetration using 0.02% Triton-X (WAKO, Japan), the samples were immersed into DAPI solution (diluted 1:1000) for 1 min. The excess staining solution was washed away using PBS. An observation of stained samples was conducted using a fluorescent microscope (Zeiss, Jena, Germany).

Masson’s trichrome stain (Muto, Tokyo, Japan) was performed to confirm collagen preservation, following standard protocols. Notably, the nuclei were first marked via immersion in Weigert’s iron hematoxylin solution for 10 min. The samples were then immersed into Orange G for 2 min, Masson’s trichrome solution for 30 min, 2.5% phosphotungstic acid for 7.5 min, and aniline blue for 15 min. Extra staining reagents were washed away using 1% acetic acid. An observation was conducted after dehydration and mounting.

Verhoeff–Van Gieson stain (Muto, Japan) was performed to confirm elastin preservation, following standard protocols. Briefly, the samples were first immersed in Maeda’s Resorsin-Fuchsin Solution for 1 h, Weigert’s iron hematoxylin solution for 10 min, and then Van Gieson’s stain solution for 10 min. An observation was conducted after dehydration and mounting.

### 2.5. Transmission Electron Microscopy (TEM)

Pieces smaller than 0.1 mm^3^ were cut from DUS samples and processed into observable slices following the standard protocol. An observation was performed using a transmission electron microscope (JEOL Ltd., Tokyo, Japan).

### 2.6. Mechanical Test

Only freshly fabricated DUS samples which underwent no freeze–thaw cycles were used in this experiment. DUS samples were longitudinally opened and clipped to a uniaxial tensile machine (Instron, Norwood, MA, USA) at the two short sides using sandpaper to increase friction (Appendix A). The tension was loaded on the samples at a constant displacement speed of 0.5 mm/min until the rupture of samples. Young’s modulus and rupture stress were calculated based on the recorded loading force, displacement, and sample sizes (Equation S1).

### 2.7. Statistical Analysis

All quantitative data were expressed in the form of means ± standard errors of the mean (SEM). Statistical differences for multiple groups were calculated using one-way ANOVA with Tukey–Kramer post hoc test. A significant difference was defined as *: *p* < 0.05; **: *p* < 0.005.

## 3. Results

### 3.1. Decellularized Uterine Scaffold (DUS) Fabrication and Characterization

Macroscopically, after pressurization and enzyme wash, DUS fabricated at 250/500/980 MPa, as shown in Figure 2a, all presented a whitish color in appearance, compared to the pink color of native uterine tissue. No obvious changes in the size or shape compared to native tissue were observed.

A qualified allograft requires the thorough removal of host cellular components. Hence, nuclear residue in DUS was first observed histologically with DAPI stain (Figure 2b) and then quantified using double-strand DNA assay (Figure 2c). In the native tissue, resident cell nuclei were stained as blue dots. The experimental group results showed that no positive stain of the nucleus was detected in all DUS groups regardless of the variation in the decellularization pressure value. The quantification results showed a significant decrease in DNA contents to 4.04% (250 MPa-DUS), 12.0% (500 MPa-DUS), and 4.14% (980 MPa-DUS) compared to native tissue (*p* < 0.005). No significant difference was found between the DUS groups. Therefore, effective decellularization was confirmed.

### 3.2. Effects of Decellularization Pressure on Structural Protein Contents in DUS

To elucidate the effect of pressure on ECM preservation, two representative structural protein components in our DUS were examined, including collagen and elastin. Histological stains and TEM observation were conducted to visualize the distribution and structure of collagen and elastin contents in DUS, and hydroxyproline and elastin assays were performed to quantify collagen and elastin contents in the DUS construct.

In Masson’s trichrome stain results (Figure 3a), native tissue includes blue stains, marking the existence of collagen fibers, red stains, marking actin filaments, and black dots, marking cell nuclei. All DUS groups showed that the blue area remained and presented a similar distribution to that of native tissue, whereas the black dots and red area were removed. The hydroxyproline assay results (Figure 3b) showed that the 250 MPa group (4195 ± 496.2 μg/g) contained a similar amount of collagen contents compared to that of native tissue (3853 ± 251.5 μg/g). However, when the decellularization pressure increased, the collagen contents in 500 MPa DUS (2656 ± 84.59 μg/g) and 980 MPa DUS (2487 ± 438.6 μg/g) decreased significantly. Notably, when comparing the DUS groups, 250 MPa DUS showed a significantly higher collagen amount than 500/980 MPa DUS. In TEM results (Figure 3c), it can be noticed that the representative triple-helical structure of collagen was preserved after decellularization regardless of pressure changes.

In the Verhoeff–Van Gieson stain results (Figure 4a), the native tissue presented black lines, indicating elastin fibers, and a red stain, indicating collagen. All DUS groups showed a similar distribution of elastin and collagen fibers to that of native tissue. The elastin assay results (Figure 4b) revealed that the number of elastin contents in the 250 MPa DUS (125.9 ± 8.321 μg/mg), 500 MPa DUS (140.4 ± 18.05 μg/mg), and 980 MPa DUS (149.7 ± 41.30 μg/mg) groups resembles that in native tissue (113.5 ± 11.67 μg/mg) as the differences were not significant.

### 3.3. Effects of Decellularization Pressure on Mechanical Properties in DUS

Adequate mechanical properties are vital for allografts. Young’s modulus and rupture strength of each DUS sample were calculated from the stress–strain curve of tensile tests, and the effect of decellularized pressure on the elasticity (Figure 5a) and strength (Figure 5b) of DUS was analyzed.

A decreasing tendency in both Young’s modulus and rupture strength of DUS was presented as the decellularization pressure increased. The average Young’s moduli were measured to be 2.3 ± 0.43 MPa in the native group, 0.94 ± 0.080 MPa in 250 MPa DUS, 0.42 ± 0.041 MPa in 500 MPa DUS, 0.38 ± 0.062 MPa in 980 MPa DUS. The measured rupture stresses for these groups were 0.27 ± 0.020 MPa in the native group, 0.20 ± 0.0093 MPa in 250 MPa DUS, 0.12 ± 0.0089 MPa in 500 MPa DUS, and 0.13 ± 0.024 MPa in 980 MPa DUS. Significant differences in Young’s modulus and rupture stress were confirmed between the 500 MPa/980 MPa groups and the native tissue. The Young’s modulus and rupture strength of the 250 MPa group seemed to differ from those of the native tissue, but no statistically significant difference was confirmed. Meanwhile, no significant difference between the DUS groups was confirmed.

## 4. Discussion

Decellularized scaffolds have been used as a conventional biomaterial to avoid organ rejection by lowering immune reactivity [27]. Although remarkable advances have been made in decellularization for the fabrication of tissue scaffolds, which have enabled tissue regeneration and organ functions, the current decellularization techniques do not offer a suitable scaffold to promote tissue regeneration due to the degradation and deconstruction of the ECM. It was previously reported that the HP method possesses the potential to fabricate decellularized uterine scaffold with more preserved ECM than the conventional SDS method [5]. Based on the previous results, the authors investigated the effect of different pressure conditions on decellularized scaffolds and developed a decellularization technique using the HP method to improve the properties of fabricated DUS.

HP treatment followed by enzyme washing removes cells and antigen epitopes from tissue. Previously, HP at 980 MPa was utilized for decellularized uterine tissue fabrication. However, excessive pressure damages structural proteins. It is suggested that the denaturation of protein induced by pressure is at first reversible under the low-pressure range, and becomes permanent when the pressure exceeds a threshold, which ranges from 200 to 600 MPa depending on protein types [26]. In some reports, the threshold pressure for collagen was considered to be 300 MPa [24,25]. Although the specific pressure value is debatable, similar conclusions that HP at 600 MPa denatures the collagen fiber structure were also reported [28,29]. Thus, a lower decellularization pressure was investigated as a potential method that could prevent damage to the ECM structure and components.

A thorough removal of resident cells in decellularized scaffolds is required to lower the risk of provoking immune rejections after implantation. Kim et al. reported that a large amount of residue DNA can still be detected after HP treatment without a wash process using DNase I [24], which indicates that the wash process may play a crucial role in DNA removal. On the other hand, cell death enables the penetration of DNase I through the nuclear envelope, which is considered necessary for the activation of chromatin breakdown [30,31]. When it comes to cell-death-inducing pressure, our previous report also showed that although HP at 50 MPa could induce cell death, it failed to remove the cell debris from the tissue, as numerous nuclei residues were detected [24]. On the other hand, it has been reported that HP at 250 MPa can kill cells in a suspension model using cultured cells [26]. However, because this report did not investigate the tissue model, the decellularization effect using 250 MPa HP remained unknown. Thus, the authors chose three representative pressure values of 250, 500, and 980 MPa to investigate the effect of HP on DUS fabricated at these pressure conditions.

In this study, all DUS samples showed the removal of blood from the tissue since a whitish color was presented compared to the pink color of native uterine tissue (Figure 2a). Furthermore, the absence of visible nucleus in DAPI stain (Figure 2b) and the significant decrease in DNA amount in DNA quantification results (Figure 2c) indicated cell removal in all DUS groups, thus highlighting the successful decellularization using a lower HP.

ECM is mainly composed of structural protein molecules, namely collagen and elastin. Collagen, one of the main components in ECM, endows the ECM with tensile strength and provides the mechanical supports required for organ function [12], while elastin provides the tissue with elasticity [32]. In this study, the authors focused on the effect of HP on collagen and elastin preservation in DUS. As shown in the collagen quantification (Figure 3a), the collagen contents in the 500 and 980 MPa groups were significantly lower than the native and 250 MPa groups. This result agrees with our hypothesis that collagen fiber was denatured due to exceeding pressure. On the other hand, the positive stain of collagen area (Figure 3b) and the existence of a triple-helical structure of collagen fiber (Figure 3c) in all DUS groups proved the superior structural preservation of the HP method regardless of pressure conditions. The existence of elastin fibers in DUS (Figure 4a) and no significant difference in elastin contents between native and DUS groups (Figure 4b) indicate that the pressure change had a negligible effect on elastin contents’ preservation.

Meanwhile, the mechanical test (Figure 5a,b) revealed that Young’s modulus and the rupture strength of the 500 and 980 MPa groups were decreased significantly compared to that of the native tissue. This result was consistent with the hydroxyproline assay results, which can be explained by the reports that the mechanical properties of ECM depend mainly on the networks of interwoven collagen and elastin fibers [15]. Relevant reports also revealed that typical cells in uterus such as endometrium stromal cells (46.8 ± 2.2 kPa) have a relatively lower Young’s modulus than collagen fibers (estimated to range from 0.1 to 0.36 GPa) [33,34]. Thus, the mechanical properties difference caused by cell absence can be neglected. Our findings also proved that excessive pressure damages the ECM components and functionality. Therefore, 250 MPa HP presented sufficient pressure for cell removal while maintaining the ECM components during DUS fabrication.

The structure and composition of ECM play an important role in the regulation of resident cell response, migration, and proliferation [27,35]. Miki et al. reported that reversed orientation of DUS led to abnormal lining of the smooth muscle layer, which indicates the importance of histological structural consistency between implanted DUS and the native tissue [16]. The topological properties, such as size and density, of collagen fiber have also been proven to be related to cell activities [36]. Therefore, the similarity in structural protein fiber amount and distribution between 250 MPa DUS and native tissue, shown in Figure 3 and Figure 4, may lead to similar cell responses, improving the regeneration outcome.

The mechanical properties of the scaffolds also affect cell response, tissue regeneration, and pregnancy outcome. Collagen network stiffness and viscoelasticity can experimentally affect single-cell and cell cluster migration dynamics [36]. Moreover, scaffold elasticity has the ability to direct stem cell differentiation [15]. Yao et al. adjusted the mechanical properties of DUS using different crosslinking conditions and determined that DUS with similar mechanical properties as native tissue yielded excellent recellularization ability [7]. Meanwhile, during gestation, the uterine tissue extends to multiple times its original size to accommodate the growing fetus, emphasizing the importance of rupture stress in DUS during gestation. The decreasing mechanical performance due to pressure increase, shown in Figure 5, indicates that 250 MPa DUS might have the best regenerative and functional potential among all the groups.

## 5. Conclusions

The aim of this project is to fabricate DUS with better preserved ECM, which is believed to be more suitable for tissue regeneration. The authors varied the pressure conditions in the HP decellularization method and studied the effect on the fabricated DUS in terms of structural protein preservation and mechanical properties. It was found that HP at 250 MPa, which is around 1/4 of the conventional conditions (980 MPa), can achieve a similar decellularization outcome as that at 980 MPa HP. Moreover, DUS fabricated under 250 MPa HP presented significantly more preserved structural proteins and mechanical properties than conventional conditions, which is promising for DUS fabrication applications.

## Figures and Tables

**Figure 1 bioengineering-10-00814-f001:**
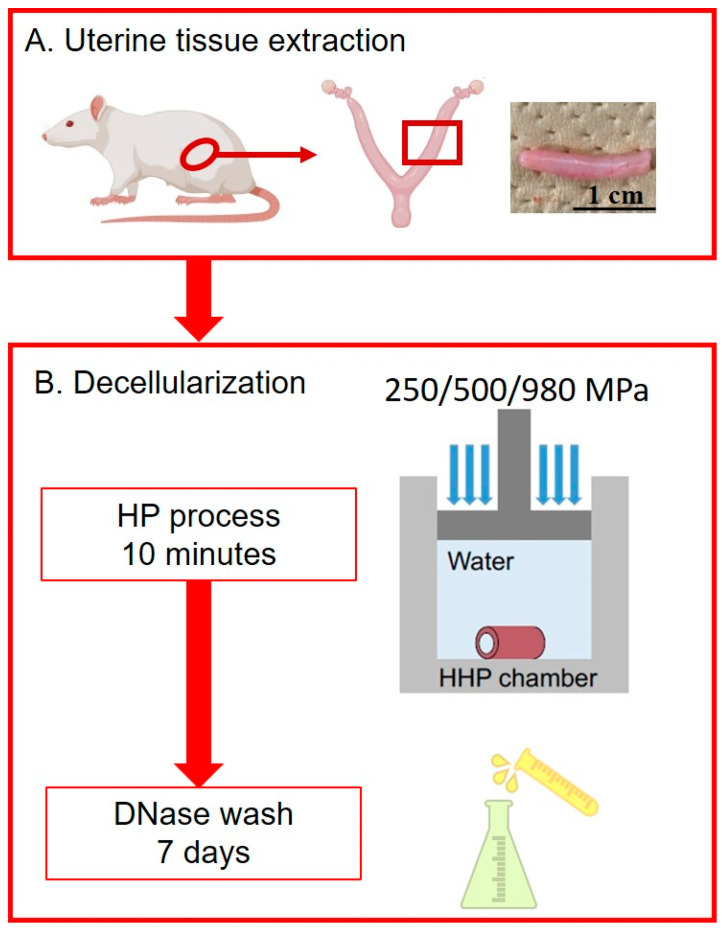
The experimental scheme.

**Figure 2 bioengineering-10-00814-f002:**
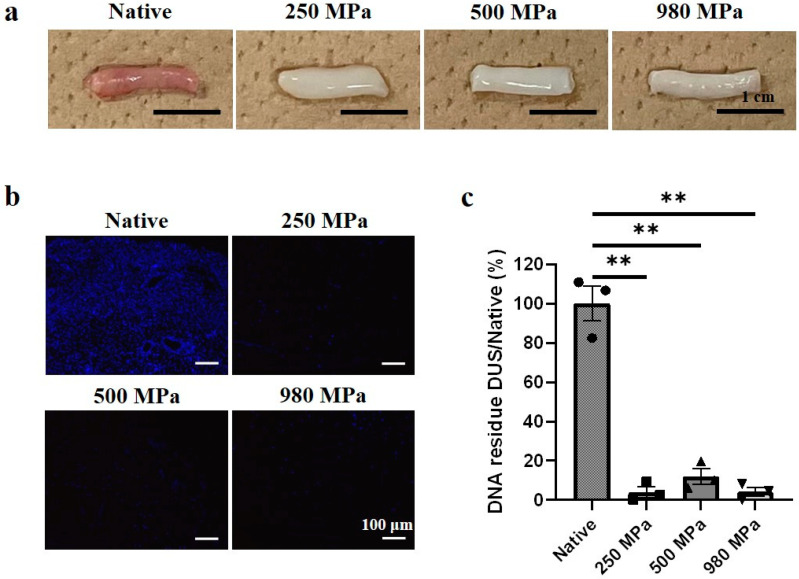
DUS fabrication and characterization. (**a**) Macroscopic observation of native uterine tissue and 250/500/980 MPa-DUS, scale bars indicate 1 cm; (**b**) DAPI stain of native uterine tissue and 250/500/980 MPa-DUS, scale bars indicate 100 μm; (**c**) DNA quantification of 250/500/980 MPa-DUS compared with native uterine tissue, represented as mean ± SEM. (*n* = 3) (** *p* < 0.005).

**Figure 3 bioengineering-10-00814-f003:**
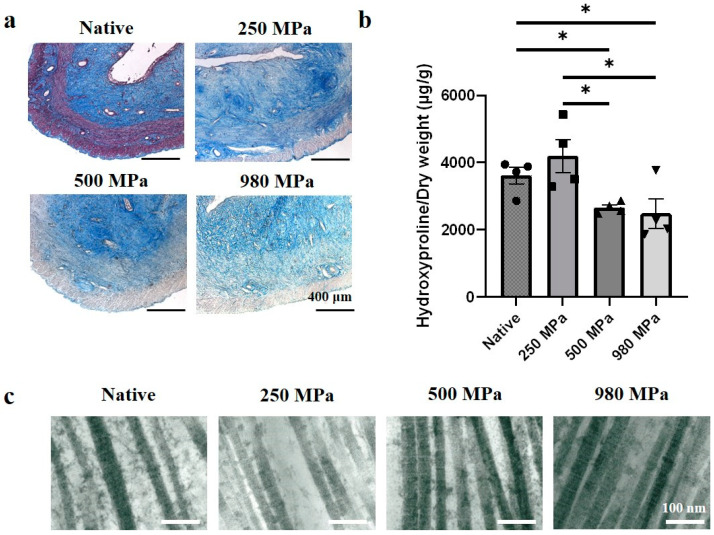
Collagen contents in DUS. (**a**) Masson’s trichrome stain of native uterine tissue and 250/500/980 MPa-DUS, scale bars indicate 400 μm; (**b**) Collagen quantification of 250/500/980 MPa-DUS compared with native uterine tissue, represented as mean ± SEM (*n* = 4) (* *p* < 0.05); (**c**) TEM image of native uterine tissue and 250/500/980 MPa-DUS, scale bars indicate 100 nm.

**Figure 4 bioengineering-10-00814-f004:**
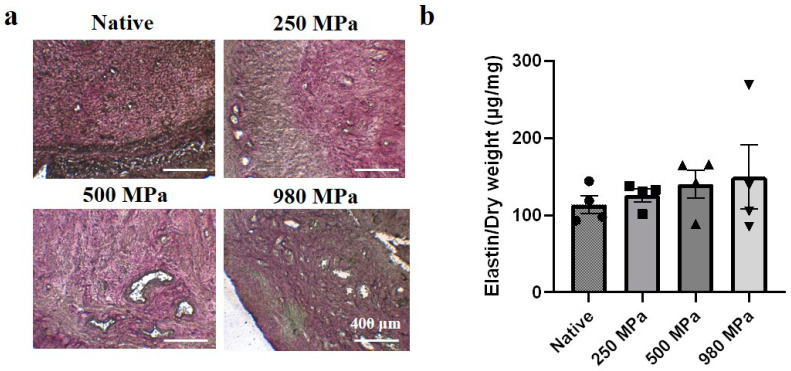
Elastin contents in DUS. (**a**) Verhoeff–Van Gieson stain of native uterine tissue and 250/500/980 MPa-DUS, scale bars indicate 400 μm; (**b**) Elastin quantification of 250/500/980 MPa-DUS compared with native uterine tissue, represented as mean ± SEM (*n* = 4).

**Figure 5 bioengineering-10-00814-f005:**
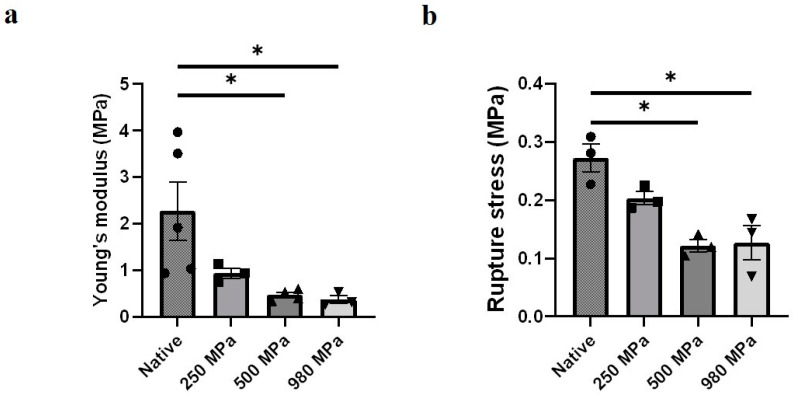
Mechanical properties of DUS. (**a**) Young’s modulus of 250/500/980 MPa-DUS compared with native uterine tissue, represented as mean ± SEM (*n* = 3) (* *p* < 0.05); (**b**) Rupture strength of 250/500/980 MPa-DUS compared with native uterine tissue, represented as mean ± SEM (*n* = 3) (* *p* < 0.05).

## Data Availability

Data are available upon request.

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
