# Peer review of "Effect of Pressure Conditions in Uterine Decellularization Using Hydrostatic Pressure on Structural Protein Preservation"

_bioengineering, 2023, doi:10.3390/bioengineering10070814_

Round 1

Reviewer 1 Report

This paper tested three pressure conditions to decellularize uterine with proper scaffold morphology. The overall research area can be significant because proper scaffolds help tissue recover morphology and functions. Previous studies use chemical or mechanical methods to decellularize varying tissues, but each protocol has limitations such as ECM damage. In this paper the authors aim to minimize such damages by lowering mechanical input. Specifically, they found that a quarter of conventional input is sufficient to decellularize uterine without significant alteration of ECM morphology. Although their current data support their argument, the Young’s modulus is highly dispersed in the control group (Native), raising a concern about reproducibility. By ignoring the error bars, the Control is two-fold stiffer than the 250MPa group, which is significant enough. In addition, it may not be that meaningful by comparing Native to other groups because the difference may result from both cells and scaffolds. Comparison between 250MPa and 500MPa/980MPa is more meaningful because these are without cells and so the only difference results from scaffolds. For all the bar charts, the author should consider displaying all data points (bean plots, for example). More importantly, it is unclear if cells in this scaffold can properly proliferate and develop morphology. The authors should add these components to be considered for publication.

Author Response

Please kindly check the attachment.

Reviewer 2 Report

In the present work, Wang et al. try to explain the effect of pressure conditions in uterine decellularization using hydrostatic pressure on structural protein preservation. There are some questions that should be considered.

1. Editing of English language and style is needed.

For example, ‘We, therefore, fabricated decellularized uterine scaffold’. In general, scientific papers are written in the third-person manner rather than the first person. Please check this throughout the manuscript.

‘15% of reproductive-aged couples suffer from infertility worldwidely’. ‘1.5 cm uterine horns were’. Numbers are generally not at the beginning of a sentence. Please check this throughout the manuscript.

Line 135, ‘0.1 mm3’, 3 should be in superscript.

Lines 195, 196 ‘(Fig.5A)’ and ‘(Fig.5B)’ should be revised.

2. Line 41, ‘achieved [5–12]. There are 8 references for this idea. Please refine these references.

3. The Introduction section is too long, and should be refined.

4. Format of references is not suitable for this Journal.

5. Some related new references are not included. For example,

Yoshimasa Y, Takao T, Katakura S, Tomisato S, Masuda H, Tanaka M, Maruyama T. A Decellularized Uterine Endometrial Scaffold Enhances Regeneration of the Endometrium in Rats. Int J Mol Sci. 2023;24(8):7605.

Masoomikarimi M, Salehi M, Noorbakhsh F, Rajaei S. A Combination of Physical and Chemical Treatments Is More Effective in The Preparation of Acellular Uterine Scaffolds. Cell J. 2023;25(1):25-34.

Moderate editing of English language required.

Author Response

Please kindly check the attachment.

Round 2

Reviewer 1 Report

My comments had been addressed properly.

Reviewer 2 Report

Thanks for author’s responses. However, format of references should been revised. Only the first letter of the title is capitalized.

Minor editing of English language required